# KLF9 Aggravates Streptozotocin-Induced Diabetic Cardiomyopathy by Inhibiting PPARγ/NRF2 Signalling

**DOI:** 10.3390/cells11213393

**Published:** 2022-10-27

**Authors:** Fangfang Li, Jingfeng Peng, Hui Feng, Yiming Yang, Jianbo Gao, Chunrui Liu, Jie Xu, Yanru Zhao, Siyu Pan, Yixiao Wang, Luhong Xu, Wenhao Qian, Jing Zong

**Affiliations:** 1Department of Cardiology, The Affiliated Hospital of Xuzhou Medical University, Xuzhou 221000, China; 2Institute of Cardiovascular Disease Research, Xuzhou Medical University, Xuzhou 221000, China; 3Department of Cardiology, The First Affiliated Hospital of Soochow University, Suzhou 215000, China

**Keywords:** diabetic cardiomyopathy, Krüppel-like Factor 9, oxidative stress, PPARγ, NRF2

## Abstract

Aims: Krüppel-like Factor 9 (KLF9) is a transcription factor that regulates multiple disease processes. Studies have focused on the role of KLF9 in the redox system. In this study, we aimed to explore the effect of KLF9 on diabetic cardiomyopathy. Methods and Results: Cardiac-specific overexpression or silencing of KLF9 in C57BL/6 J mice was induced with an adeno-associated virus 9 (AAV9) delivery system. Mice were also subjected to streptozotocin injection to establish a diabetic cardiomyopathy model. In addition, neonatal rat cardiomyocytes were used to assess the possible role of KLF9 in vitro by incubation with KLF9 adenovirus or small interfering RNA against KLF9. To clarify the involvement of peroxisome proliferator-activated receptors (PPARγ), mice were subjected to GW9662 injection to inhibit PPARγ. KLF9 was upregulated in the hearts of mice with diabetic cardiomyopathy and in cardiomyocytes. In addition, KLF9 overexpression in the heart deteriorated cardiac function and aggravated hypertrophic fibrosis, the inflammatory response and oxidative stress in mice with diabetic cardiomyopathy. Conversely, cardiac-specific silencing of KLF9 ameliorated cardiac dysfunction and alleviated hypertrophy, fibrosis, the cardiac inflammatory response and oxidative stress. In vitro, KLF9 silencing in cardiomyocytes enhanced inflammatory cytokine release and oxidative stress; KLF9 overexpression increased these detrimental responses. Moreover, KLF9 was found to regulate the transcription of PPARγ, which suppressed the expression and nuclear translocation of nuclear Factor E2-related Factor 2 (NRF2). In mice injected with a PPARγ inhibitor, the protective effects of KLF9 knockdown on diabetic cardiomyopathy were counteracted by GW9662 injection. Conclusions: KLF9 aggravates cardiac dysfunction, the inflammatory response and oxidative stress in mice with diabetic cardiomyopathy. KLF9 may become a therapeutic target for diabetic cardiomyopathy.

## 1. Introduction

Reactive oxygen species (ROS) act as second messengers under physiological conditions. However, when the production of ROS exceeds the capacity of the antioxidant system, it causes an imbalance in the redox system, leading to oxidative stress [1,2]. In individuals with type 1 diabetes, diabetic cardiomyopathy (DCM) is characterized by hypoinsulinaemia and hyperglycaemia, while in individuals with type 2 diabetes, hyperinsulinemia or insulin resistance are the main features of diabetic cardiomyopathy [2]. In both type 1 and type 2 diabetes, diabetic cardiomyopathy is characterized by excessive inflammatory responses and oxidative stress. Glucose autoxidation, protein glycosylation, and oxidative degradation of glycosylated proteins induced by hyperglycaemia lead to the enhanced production of ROS under diabetic conditions [3]. These factors accelerate the impairment of cardiomyocytes, which in turn magnifies the inflammatory response and ultimately leads to cardiac dysfunction [4]. Many studies have suggested that inhibiting oxidative stress can suppress the DCM process [5,6,7]. However, results from these studies were generated in animal models and, as such, could not be translated to clinical application temporarily.

As a redox regulator, NF-E2-related Factor 2 (NRF2) can regulate various molecules that participate in redox balance, including glutathione peroxidase (GPX), superoxide dismutase (SOD), and haeme oxygenase-1 (HO-1). Our previous study found that NRF2 was downregulated in DCM hearts and that the DCM process was suppressed by increasing the level of NRF2 [8]. NRF2 can be regulated by many associated factors, such as sirtuins, ATF4 and peroxisome proliferator-activated receptors (PPARγ) [9]. Thus, a therapeutic method for regulating NRF2 may be a promising approach to inhibit the DCM process.

KLF9 is a transcription factor of the Krüppel-like family [10]. KLF9 was also reported to be an essential factor involved in the redox system [11]. Zucker SN et al. found that KLF9 acted as a pro-oxidative stress factor that can be activated by NRF2 in response to severe mitochondrial ROS [12]. However, Parga JA found that KLF9 caused a reduction in the production of ROS induced by angiotensin II in neuronal cells and improved their survival [13]. Furthermore, KLF9 plays a critical role in NRF2-mediated antioxidative effects in trioxide-induced hepatotoxicity in rats [14]. These contradictory results indicate the associated effects of KLF9 on redox balance in cardiovascular disease, especially DCM. In this study, we overexpressed and knocked down KLF9 in the heart to explore the role of KLF9 in DCM.

## 2. Methods

### 2.1. Animals

Male C57BL/6 J mice (8 weeks old) were purchased from the Chinese Academy of Medical Sciences (Beijing, China). All animal experiments were approved by the Animal Care and Use Committee of the First Affiliated Hospital of Xuzhou Medical University (protocol number: 00019864). To overexpress KLF9 in mice, a single injection of adeno-associated virus 9 (AAV9)-KLF9 was administered 10 weeks after the final streptozotocin (STZ) injection (*n* = 15). To knock down KLF9 in mice, a single injection of AAV9-shKLF9 was administered 10 weeks after the final streptozotocin (STZ) injection (*n* = 15). Control mice were administered the same gene copy number of either AAV9-negative control (NC) or AAV9-shRNA (*n* = 15 mice per group). To inhibit PPARγ, mice were treated with a 50 μg/kg daily dose of GW9662 for 28 days at 12 weeks after the final STZ injection (*n* = 15 mice per group).

### 2.2. Animal Model

The diabetes model was established by intraperitoneal injection of STZ (50 mg/kg for 5 consecutive days) [8]. Control mice were injected with the same volume of vehicle solution (0.1 mol/L citrate buffer, pH 4.5). Fasting blood glucose levels were evaluated 7 days after the final injection. A fasting blood glucose level greater than 16.6 mmol/L in three independent tests was considered to indicate successful establishment of the diabetes model. Ten weeks after the final STZ injection, mice were injected with AAV9-KLF9 or AAV9-shKLF9. Blood glucose levels and body weight data were collected every month. Sixteen weeks after the final STZ injection, the mice were sacrificed, and heart samples were collected.

### 2.3. AAV9 Construction and Delivery

AAV9-KLF9 (control: AAV9-NC) and AAV9-shKLF9 (control: AAV9-shKLF9) were generated by Vigene Bioscience Company (Jinan, China) under the cTnT promoter. Ten weeks after STZ injection, mice were subjected to AAV9 injection (retroorbital venous plexus, 60 μL, 6.5–7.5 × 10^13^ gene copies/mL per mouse), as described in our previous study [8].

### 2.4. Echocardiography and Pressure-Volume Loop Evaluation

Transthoracic echocardiography was performed as previously described [15]. Isoflurane (1.5%) was used to anaesthetize the mice. Echocardiography was performed with a 10-MHz linear array ultrasound transducer. An M shape was observed on the echocardiogram. The left ventricular (LV) end diastolic dimension (LVEDd) and end systolic dimension (LVESd) were determined. The LV ejection fraction (LVEF), LV fractional shortening (LVFS) and E/A ratio were calculated. The LV fraction shortening (LVFS) was calculated using the following formula: LVFS(%) = (LVEDd − LVESd)/LVEDd × 100%. A total of 10 mice per group were subjected to transthoracic echocardiography. Diastolic function was assessed by the ratio of early diastolic peak (E) to late diastolic peak (A) of mitral valve blood flow as measured by pulse Doppler ultrasound.

For pressure-volume loop measurement, mice were anaesthetized with 1.5% isoflurane. A microtip pressure-volume catheter (SPR-839; Millar Instruments, Houston, TX, USA) was inserted into the right carotid artery and then into the left ventricle. After stabilization, a Millar Pressure-Volume System (MPVS-400; Millar Instruments) was used to record pressure and volume signals. PVAN data analysis software was used to analyse the data.

### 2.5. ELISA Detection of Inflammatory Cytokines

Tumour necrosis factor α, interleukin (IL)-1, and IL-6 in mouse hearts as well as cardiomyocytes and macrophages were detected with ELISA kits purchased from BioLegend (430901, 432604, 431304). An ELISA plate reader (Synergy HT, BioTek, Winooski, VT, USA) was used to determine the fluorescence intensity values.

### 2.6. Immunohistochemical Staining

After dehydration, we used the heat-mediated antigen retrieval method as referred to in our previous study [16], and blocking with 8% goat serum; hearts were incubated first with an anti-CD68 antibody (Abcam, 1:100 dilution) and then with an anti-rabbit HRP-conjugated secondary antibody (Gene Tech, Shanghai, China). A peroxide-based substrate DAB kit (Gene Tech, Shanghai, China) was used for colour development.

### 2.7. Oxidative Stress Assay

Dihydroethidium (DHE) was used to assess the production of intracellular superoxide anions (O_2_^−^). Briefly, freshly obtained heart samples were embedded in Tissue-Tek OCT Compound Histomount and frozen on dry ice. Samples were incubated with superoxide dismutase-polyethylene glycol (SOD-PEG) (Sigma-Aldrich) and stained with DHE (50 μL per sample) (AnaSpec Inc., Fremont, CA, USA) at 37 °C for 30 min to confirm the absence of autofluorescence. After washing, the tissues were covered with 70% glycerol.

The ROS level, superoxide dismutase 2 (SOD2) activity, nicotinamide adenine dinucleotide phosphate (NADPH) oxidase activity, glutathione peroxidase (Gpx) activity and malondialdehyde (MDA) level in heart tissue and cardiomyocytes were detected with the corresponding kits purchased from Beyotime (Shanghai, China) and according to the manufacturer’s instructions.

### 2.8. Cardiomyocyte Isolation and Culture

A neonatal rat cardiomyocyte (NRCM) culture was performed as previously described [15]. Briefly, the hearts were quickly removed from 1- to 3-day-old Sprague–Dawley rats, and the ventricles were preserved and digested with 0.125% trypsin-EDTA (Gibco) four times for 15 min each. Digestion was stopped in DMEM-F12 supplemented with 15% foetal bovine serum (FBS) (Gibco, USA). After five digestion steps, the cells were collected and incubated in a 100-mm dish in DMEM-F12 supplemented with 15% FBS. After 90 min, the cell culture medium was collected, and NRCMs in the upper layer of the cell culture medium were seeded into a 6-well plate. This step was performed to remove the noncardiac myocytes adhering to the bottom of the 100-mm dish. NRCMs were identified by α-actin staining. Cells were exposed to high glucose (HG, 33.3 mM) to establish the in vitro DCM model. Control cells were exposed to normal glucose (5.5 mM; 27.5 mM mannitol was used to maintain the same osmotic pressure as the HG group). To overexpress KLF9, cells were transfected with Ad-KLF9 (MOI = 100). To knock down KLF9, cells were transfected with KLF9 siRNA (MOI = 100, Vibo Biotech) using the Lipo6000 transfection reagent (Beyotime, Nantong 226000, China). Cells were also treated with GW9662 (10 μM, MedChemExpress, Princeton, NJ 08544, USA) to inhibit PPARγ or with ML385 (1 μM, MedChemExpress) to inhibit NRF2.

### 2.9. Western Blotting and qPCR

Total protein was isolated from heart tissue and NRCMs for whole cell lysates. As for nuclear NRF2 and Histone H3 expression, nuclear proteins were extracted. The proteins were then subjected to SDS–PAGE (50 μg per sample). After transfer onto Immobilon membranes (Millipore, Billerica, MA, USA), the proteins were incubated with primary antibodies overnight at 4 °C. The primary antibodies included anti-KLF9 (A7196; ABclonal), anti-NRF2, anti-SIRT1 (all purchased from Abcam, 1:1000 dilution), anti-Histone H3 and anti-GAPDH (both purchased from Cell Signaling Technology, 1:1000 dilution). Blots were developed with enhanced chemiluminescence (ECL) reagents (Bio-Rad, Hercules, CA, USA) and imaged with a ChemiDoc MP Imaging System (Bio-Rad). GAPDH was used as the internal reference protein. Each band represents a sample of a heart/a group of cells. Image Lab software was used for optical density value analysis. After GAPDH was used for standardization, the data are expressed in the form of mean plus minus standard deviation.

Total mRNA was extracted using TRIzol reagent (#15596-026, Invitrogen, Waltham, MA, USA) according to the manufacturer’s instructions, as previously described [8]. The primers used are listed in Table 1.

### 2.10. Luciferase Assay

NRCMs were transfected with 500 ng of PPRE3-luciferase reporter plasmid or NRF2-luciferase reporter plasmid in combination with Ad-KLF9 or KLF9 siRNA. After cells were harvested and washed, they were lysed in a passive lysis buffer (Promega, Madison, WI 53711, USA). A GloMax^®^ 20/20 Luminometer (Promega) was used to detect luciferase.

### 2.11. Statistical Analysis

All data are expressed as the mean ± SD values. Differences among groups were analyzed by two-way analysis of variance followed by Tukey’s post-hoc test. Comparisons between two groups were analysed by unpaired Student’s *t*-test. Statistical significance was assumed when a *p* value was less than 0.05.

## 3. Results

### 3.1. KLF9 Is Upregulated in DCM

We first assessed the alteration of KLF9 expression in DCM. As expected, the expression of KLF9 was upregulated during DCM in mouse hearts (Figure 1A,B). We then assessed the level of KLF9 in cardiomyocytes. Consistent with the above findings, KLF9 was increased in cardiomyocytes exposed to high glucose for 48 h (Figure 1C,D). This finding indicates that KLF9 participates in the pathology of DCM.

### 3.2. KLF9 Overexpression Suppresses Cardiac Dysfunction in DCM

To evaluate the role of KLF9 in DCM, we used an AAV9 system to overexpress KLF9 in mouse hearts. Six weeks after AAV9-KLF9 injection, the level of KLF9 was much higher in the KLF9 overexpression group than in the control group (Figure 2A). Immunofluorescence staining also confirmed that KLF9 was mainly localized in cardiomyocytes (Figure 2B). During the 16-week follow-up period, blood glucose was higher in DCM mice than in mice in the control group, but no significant difference was found between the two DCM groups (Figure 2C). The body weight of mice dramatically decreased in the DCM group, but no significant difference was observed between the two DCM groups (Figure 2D). Cardiac function was assessed by two methods: echocardiography and pressure-volume loop evaluation. As shown in Appendix A, mice in the AAV9-KLF9-DCM group showed a larger LVESd and a lower LVEF, LVFS and E/A ratio than mice in the AAV9-NC-DCM group. In addition, the pressure-volume loop evaluation results showed that cardiac output and the maximum rates of left ventricular pressure increase and decay (dt/dp max and dt/dp min, respectively) were reduced in the AAV9-KLF9-DCM group compared with the AAV9-NC-DCM group. Moreover, the Tau value was higher in the AAV9-KLF9-DCM group than in the NC-DCM group (Appendix A and Figure 2B). These data suggest that KLF9 aggravates cardiac dysfunction in DCM. We also assessed cardiac hypertrophy and fibrosis in mouse hearts. H&E staining showed an increased cell surface area in the hearts of DCM mice; PSR staining also confirmed a significant fibrotic response in DCM mice (Figure 2E,F). KLF9 overexpression increased the hypertrophic and fibrotic responses, as evidenced by the augmented cell surface area and collagen volume as well as the altered mRNA levels of related genes (β-MHC, α-MHC, collagen I, and collagen III) (Figure 2E–G).

### 3.3. KLF9 Exacerbates Inflammation and Oxidative Stress in DCM

Inflammation and oxidative stress are the main features of DCM. We assessed the role of KLF9 in mediating these pathological characteristics. Macrophages were stained for CD68, a known molecular marker. The number of CD68-positive cells was higher in the AAV9-KLF9-DCM group than in the NC-DCM group (Figure 3A). In addition, the release of inflammatory cytokines was assessed by ELISA. In heart tissue from DCM mice, the release of TNFα, IL-1 and IL-6 was enhanced compared with that in control mice. KLF9 overexpression dramatically increased the levels of these proinflammatory factors (Figure 3B). The superoxide anion level was assessed by DHE staining. As shown in Figure 3C, the level of superoxide anion was much higher in the AAV9-KLF9-DCM group than in the NC-DCM group. Consistent with these findings, the level of MDA, an intermediate product of lipid metabolism, was also higher in the AAV9-KLF9-DCM group than in the NC-DCM group. We also detected oxidase and antioxidant activities. NADPH activity was increased, but SOD2 and Gpx activity was reduced in AAV9-KLF9-DCM mouse hearts compared with NC-DCM mouse hearts.

### 3.4. KLF9 Knockdown Ameliorates Cardiac Dysfunction in DCM

Whether KLF9 knockdown ameliorates the process of DCM is unknown. We knocked down KLF9 via the AAV9 system. The level of KLF9 was sharply reduced six weeks after AAV9-shKLF9 injection (Figure 4A). This reduction in KLF9 was mainly observed in cardiomyocytes, as confirmed by immunofluorescence staining (Figure 4B). Blood glucose was increased in the DCM groups (Figure 4C). The body weight in the DCM group was reduced gradually compared with that in the control group. Under DCM conditions, the body weight was slightly reduced in the AAV9-shKLF9 group compared with the AAV9-shRNA group (Figure 4D). Cardiac function parameters are shown in Appendix A; the LVESd was smaller in mice subjected to AAV9-shKLF9 injection than in mice injected to AAV9-shRNA. The LVEF, LVFS and E/A ratio were increased in mice subjected to AAV9-shKLF9 injection. No difference was found in the heart rate among the four groups. Compared with the AAV9-shRNA group, the AAV9-shKLF9 group exhibited increased cardiac output, dt/dp max, and dt/dp min and reduced Tau (Appendix A). Cardiac hypertrophy and fibrosis were ameliorated in mice receiving an AAV9-shKLF9 injection, as shown by the decreased cell surface area and collagen volume and increased alterations in hypertrophy and fibrosis markers (Figure 4E–G).

### 3.5. KLF9 Knockdown Suppressed Inflammation and Oxidative Stress in DCM

The number of CD68-positive cells was lower in the KLF9 knockdown mice than in control DCM mice (Figure 5A). The release of TNFα, IL-1 and IL-6 was also markedly decreased in KLF9 knockdown mice under DCM conditions (Figure 5B). The levels of superoxide anion and MDA in heart tissue were sharply decreased in mice injected with AAV9-shKLF9 under DCM conditions. Regarding oxidase and antioxidant activities, the activity of NADPH was decreased and that of SOD2 and Gpx was increased in the hearts of mice in the AAV9-shKLF9 group compared with the hearts of mice in the AAV9-shRNA group (Figure 5E).

### 3.6. KLF9 Affects Cardiomyocytes In Vitro

NRCMs were isolated and cultured to explore the direct role of KLF9 in cardiomyocytes. Cells were transfected with Ad-KLF9 or KLF9 siRNA to overexpress or knock down KLF9 (Figure 6A). Eight hours after transfection, cells were exposed to HG for 48 h. The levels of TNFα, IL-1 and IL-6 were assessed by ELISA. As shown in Figure 6B, HG increased the release of TNFα, IL-1 and IL-6, while KLF9 silencing inhibited the release of these cytokines. Oxidative stress was also assessed. The levels of ROS and MDA and the activity of NADPH increased under HG stimulation, while KLF9 silencing reduced the ROS and MDA levels and decreased NADPH activity (Figure 6C). The activity of SOD2 and Gpx was decreased in the KLF9-silenced group (Figure 6D). Conversely, KLF9 overexpression increased the release of these proinflammatory cytokines (Figure 6E), increased the ROS level (Figure 6F), and accelerated the imbalance in the oxidation-antioxidant system (Figure 6G). This evidence indicates that KLF9 plays a direct role in cardiomyocytes.

### 3.7. KLF9 Regulates PPARγ/NRF2 Signalling

We sought to identify the signalling pathway that may be the target of KLF9 in cardiomyocytes. As a result, we found that the NRF2 level was decreased in mouse hearts overexpressing KLF9. The nuclear level of NRF2 was also decreased in AAV9-KLF9 mouse hearts (Figure 7A). The total and nuclear levels of NRF2 were enhanced in cells transfected with KLF9 siRNA (Figure 7B). We also found that upstream of NRF2, PPARγ expression was decreased by KLF9 overexpression but enhanced by KLF9 knockdown (Figure 7A,B). We then explored the relationship between KLF9 and PPARγ/NRF2. We observed a significant decrease in PPARγ-Luc transcriptional activity in KLF9-overexpressing cells compared with control cells (Figure 7C). KLF9 knockdown increased PPARγ-Luc transcriptional activity (Figure 7C). However, NRF2-Luc transcriptional activity was not changed by KLF9 siRNA or Ad-KLF9 treatment (Figure 7D). These data indicate that KLF9 regulates NRF2 expression by targeting PPARγ transcription.

### 3.8. PPARγ/NRF2 Inhibition Counteracted The Protective Effects of KLF9 Knockdown

We sought to confirm the requirement for PPARγ/NRF2 in the protective role of KLF9 knockdown in cardiomyocytes. Cells were treated with the PPARγ inhibitor GW9662 (10 μM) or an NRF2 inhibitor (ML385, 1 μM) for 12 h. The inflammatory response and oxidative stress were assessed. Cells treated with GW9662 or ML385 alone under HG conditions revealed an increased inflammatory response and excessive oxidative stress (Figure 8A–D). Moreover, in KLF9-silenced cells treated with GW9662 or ML385, the inflammatory response and oxidative stress level showed no difference between the GW9662 and ML385 groups (Figure 8A–D). Thus, these data confirm that KLF9 relies on PPARγ/NRF2 to exert its cardiomyocyte-protective effects.

### 3.9. Cardiac Dysfunction Persists in KLF9 Knockdown PPARγ Inhibited Mice

Mice injected with GW9662 were used to confirm the functional role of KLF9 in PPARγ/NRF2 signalling in vivo. Under DCM conditions, blood glucose levels were higher in both the GW9662-DCM group and shKLF9+ GW9662 group than in the DCM-VEH group. Body weight was reduced in these three DCM groups compared with that in the non-DCM groups. No difference was observed in body weight among these three DCM groups (Figure 9B). Regarding cardiac function, deteriorating systolic and diastolic function was observed in GW9662-DCM mice compared with DCM-VEH mice (Appendix A). However, in mice with both GW9662 injection and KLF9 knockdown, the deteriorated cardiac dysfunction was not different from that in mice with only GW9662 injection (Appendix A). Cardiac inflammation was also assessed in the AAV9-KLF9+ GW9662 group. The number of CD68-positive cells and amount of proinflammatory cytokines released in heart tissue were higher in the GW9662-DCM group than in the DCM-VEH group. However, no difference was observed in cardiac inflammation between GW9662-DCM mice and shKLF9+ GW9662 mice (Figure 9B,C). Oxidative stress and the imbalance in the redox system were increased in GW9662-DCM mice compared with DCM-VEH mice. However, KLF9 knockdown did not exert a protective effect in mice injected with GW9662 [17] under DCM conditions (Figure 9D–F).

## 4. Discussion

KLF9 regulates many important biological processes, such as proliferation, apoptosis, differentiation, and development [17]. As a transcription factor, KLF9 was initially found to regulate adipogenesis [18]. In this study, we first identified the role of KLF9 in a cardiovascular disease: diabetic cardiomyopathy. We found that KLF9 was upregulated during the pathological process of DCM. Second, we found that KLF9 acted as a cardiac-deteriorating effector during DCM, since when we manipulated the expression level of KLF9, DCM progression was suppressed with low levels of KLF9 but exacerbated with high levels of KLF9. These findings may be beneficial for considering the delivery or regulation of KLF9 in DCM treatment.

DCM is a disease that is defined as cardiac dysfunction independent of hypertension, coronary heart disease, cardiac ischaemia and other cardiomyopathies [19]. It is a disease resulting from diabetes with metabolic dysregulation, especially hyperglycaemia and hyperlipidemia. Advanced glycation end products (AGEs) cause protein, lipid and nucleotide glycosylation in cardiomyocytes, leading to cardiomyocyte dysfunction and death [20]. Chronic DCM also involves cardiac inflammation, fibrosis and cardiac dysfunction [20]. Yan Q et al. reported that KLF9 aggravates ischaemic injury in cardiomyocytes [21]. In our study, we found an increase in the level of KLF9 in mouse hearts with the DCM phenotype. The overexpression of KLF9 in mouse hearts resulted in an exacerbated phenotype. The role of KLF9 might be mediated through the regulation of cardiac metabolism. However, we did not find a difference in body weight or blood glucose levels between the KLF9 overexpression group and the vehicle group during the 16-week period of DCM progression. This finding indicates that the deleterious effects of KLF9 did not rely on metabolic regulation. However, in the KLF9 knockdown experiment, we observed a slight decrease in mice body weight. These results may be attributed to the effect of KLF9 on PPARγ.

In our in vivo study, we found deleterious effects of KLF9 on inflammation and oxidative stress. These two characteristics are the main features and molecular mechanisms leading to cardiac dysfunction during DCM. In an in vitro study, KLF9 was shown to act directly on cardiomyocytes and showed the same proinflammatory and antioxidative stress effects reported here. Previous studies also reported the pro-oxidative stress effect of KLF9. Yan Q found that KLF9 aggravated ischemic injury in cardiomyocytes through augmenting oxidative stress [21]. KLF9 increased ROS regulating melanoma progression in a stage-specific manner [22]. By promoting oxidative stress and inflammation in mice fed a high-fat Diet, KLF9 suppresses hepatocellular carcinoma progress [23]. KLF9 was also reported to be associated with inflammation cytokines release. Knockout of KLF9 ameliorates LPS induced acute lung injury and inflammation in mice [24]. Many studies have shown that oxidative stress mediated by excessive ROS levels is the pathogenic mechanism underlying both type 1 and type 2 diabetes-associated cardiomyopathy [5,8,25]. This excessive oxidative stress weakens the cytoplasmic membrane and sarcoplasmic reticulum function, leading to myofibrillar abnormalities [4].

In an in vitro study, we found that KLF9 can decrease the expression and nuclear translocation of NRF2. NRF2 is a transcription factor that regulates key antioxidant cellular responses [26]. Once activated by upstream molecules, NRF2 translocates to the nucleus and binds to the promoter region of many antioxidases, such as SOD, thioredoxin peroxidase, Gpx and catalase [9]. In addition to its transcription activity, NRF2 was also proved to anti-NF-kapaB inflammation pathway Nrf2 appears to spark cell survival signaling, preserve mitochondrial biogenesis and regulating autophagy, which exert protective effects on many cardiovascular diseases [26]. We then found that KLF9 inhibited PPARγ expression and directly reduced the transcription of PPARγ. PPARγ is a nuclear hormone receptor that regulates the homeostasis of glucose and lipids and NRF2 activation [8]. Studies have reported that KLF9 reduces the ROS level in neuronal cells and improves survival [13]. However, Yan Q et al. reported that KLF9 aggravated ischaemic injury in cardiomyocytes by undermining Txnrd2-mediated ROS clearance [21]. These inconsistencies may result from differences in cell types or disease statuses. However, in our study, KLF9 played a deleterious role, which is consistent with the results of Yan Q’s study. In adipocytes, KLF9 was found to activate PPARγ expression, which promoted adipogenesis [18]. It was also found that under serious oxidative stress, NRF2 could activate KLF9, which subsequently inhibited the transcription of antioxidant genes [9]. Moreover, KLF9 could not regulate NRF2 expression [9]. In our study, we found a reduced expression of NRF2 in DCM mouse hearts. KLF9 overexpression inhibited the NRF2 protein level. Our luciferase assay confirmed that KLF9 regulates the expression of NRF2 by inhibiting PPARγ. When we treated mice with a PPARγ inhibitor, the protective effect of KLF9 knockdown on DCM was counteracted. These results suggest that the PPARγ-NRF2-oxidative stress pathway is the target of KLF9 in DCM pathology.

## 5. Conclusions

Taken together, our findings indicate that KLF9 can exacerbate cardiac inflammation, reduce oxidative stress and deteriorate cardiac dysfunction. KLF9 exerts a negative effect by acting directly on cardiomyocytes. KLF9 regulates the PPARγ-NRF2-oxidative stress pathway in cardiomyocytes. Translational clinical research on the therapeutic effects of KLF9 in DCM is promising.

## Figures and Tables

**Figure 1 cells-11-03393-f001:**
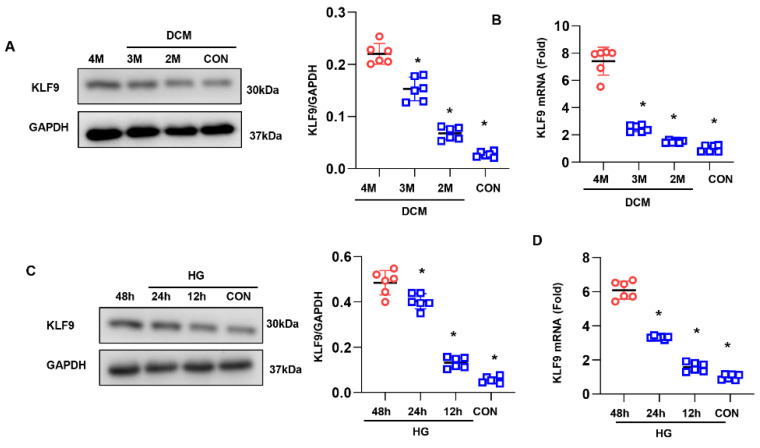
KLF9 is upregulated in DCM: The protein (**A**) and mRNA (**B**) levels of KLF9 in the hearts of 2-, 3-, and 4-month-old mice after the final STZ injection (*n* = 6). The protein (**C**) and mRNA (**D**) levels of KLF9 at 12, 24, and 48 h in cardiomyocytes exposed to high glucose (HG) (*n* = 6). * *p* < 0.05 vs. the CON group.

**Figure 2 cells-11-03393-f002:**
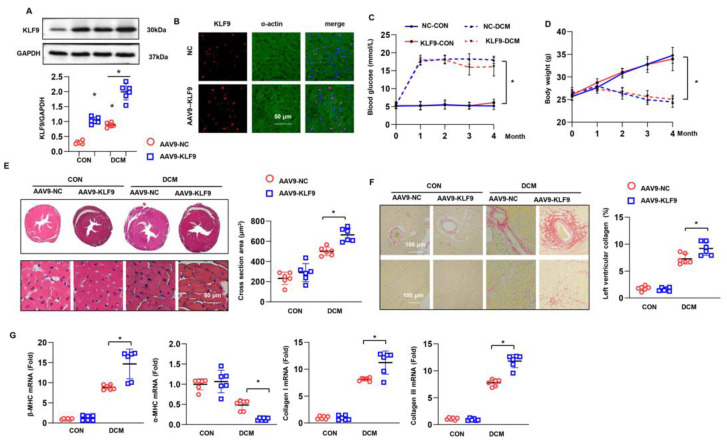
KLF9 overexpression aggravates cardiac dysfunction in DCM: (**A**) The protein level of KLF9 in mice 6 weeks after AAV9-KLF9 injection (*n* = 6). * *p* < 0.05 vs. KLF9 0 w. (**B**) Immunofluorescence staining of KLF9 and α-actin in mouse hearts 6 weeks after AAV9-KLF9 injection (*n* = 5). (**C**) Blood glucose in DCM mice at the indicated time points (*n* = 12). (**D**) The body weight of DCM mice at the indicated time points (*n* = 12). (**E**) H&E staining and cross-sectional area results (*n* = 6). (**F**) PSR staining and collagen volume results (*n* = 6). (**G**) The mRNA levels of hypertrophy and fibrosis markers in each group (*n* = 6). * *p* < 0.05.

**Figure 3 cells-11-03393-f003:**
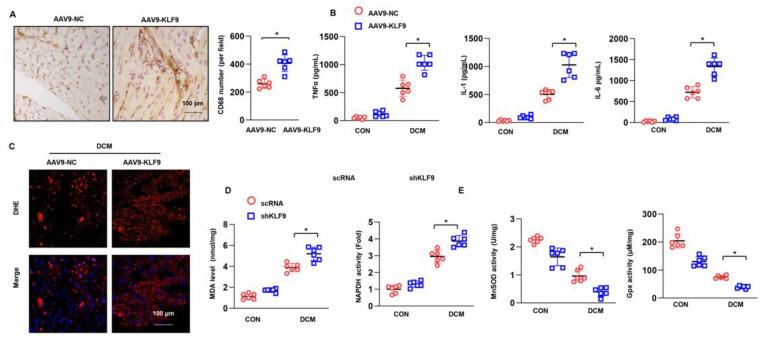
KLF9 overexpression aggravates inflammation and oxidative stress in DCM. (**A**) CD68 staining and quantitative results in DCM mice receiving AAV9-KLF9 injection (*n* = 5). (**B**) Cytokine levels in heart tissue in DCM mice receiving AAV9-KLF9 injection (*n* = 6). (**C**) DHE staining in DCM mice receiving AAV9-KLF9 injection (*n* = 5). (**D**) MDA levels and NADPH oxidase activity in DCM mice receiving AAV9-KLF9 injection (*n* = 6). (**E**) SOD2 and Gpx activity (*n* = 6). * *p* < 0.05.

**Figure 4 cells-11-03393-f004:**
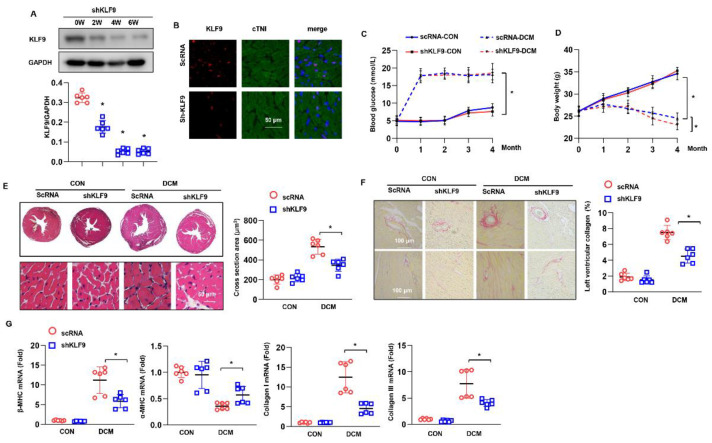
KLF9 knockdown suppresses cardiac dysfunction in DCM. (**A**) The protein level of KLF9 in mice receiving AAV9-shKLF9 injection (*n* = 6). (**B**) Immunofluorescence staining of KLF9 and α-actin in mouse hearts 6 weeks after AAV9-shKLF9 injection (*n* = 5). (**C**) Blood glucose in DCM mice at the indicated time points (*n* = 12). (**D**) The body weight of DCM mice at the indicated time points (*n* = 12). (**E**) H&E staining and cross-sectional area results (*n* = 6). (**F**) PSR staining and collagen volume results (*n* = 6). (**G**) The mRNA levels of hypertrophy and fibrosis markers in each group (*n* = 6). * *p* < 0.05.

**Figure 5 cells-11-03393-f005:**
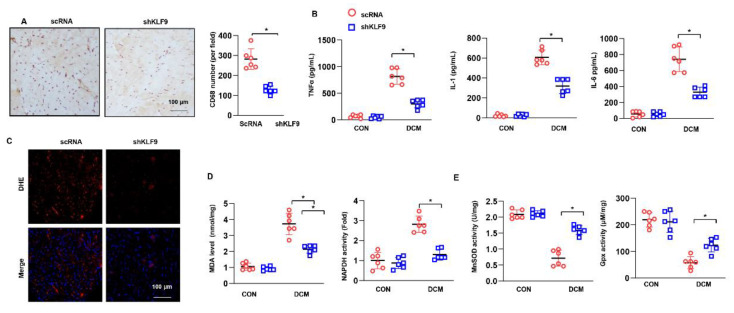
KLF9 knockdown ameliorates inflammation and oxidative stress in DCM. (**A**) CD68 staining and quantitative results in DCM mice receiving AAV9-shKLF9 injection (*n* = 5). (**B**) Cytokine levels in heart tissue in DCM mice receiving AAV9-shKLF9 injection (*n* = 6). (**C**) DHE staining in DCM mice receiving AAV9-shKLF9 injection (*n* = 5). (**D**) MDA levels and NADPH oxidase activity in DCM mice receiving AAV9-shKLF9 injection (*n* = 6). (**E**) SOD2 and Gpx activity (*n* = 6). * *p* < 0.05.

**Figure 6 cells-11-03393-f006:**
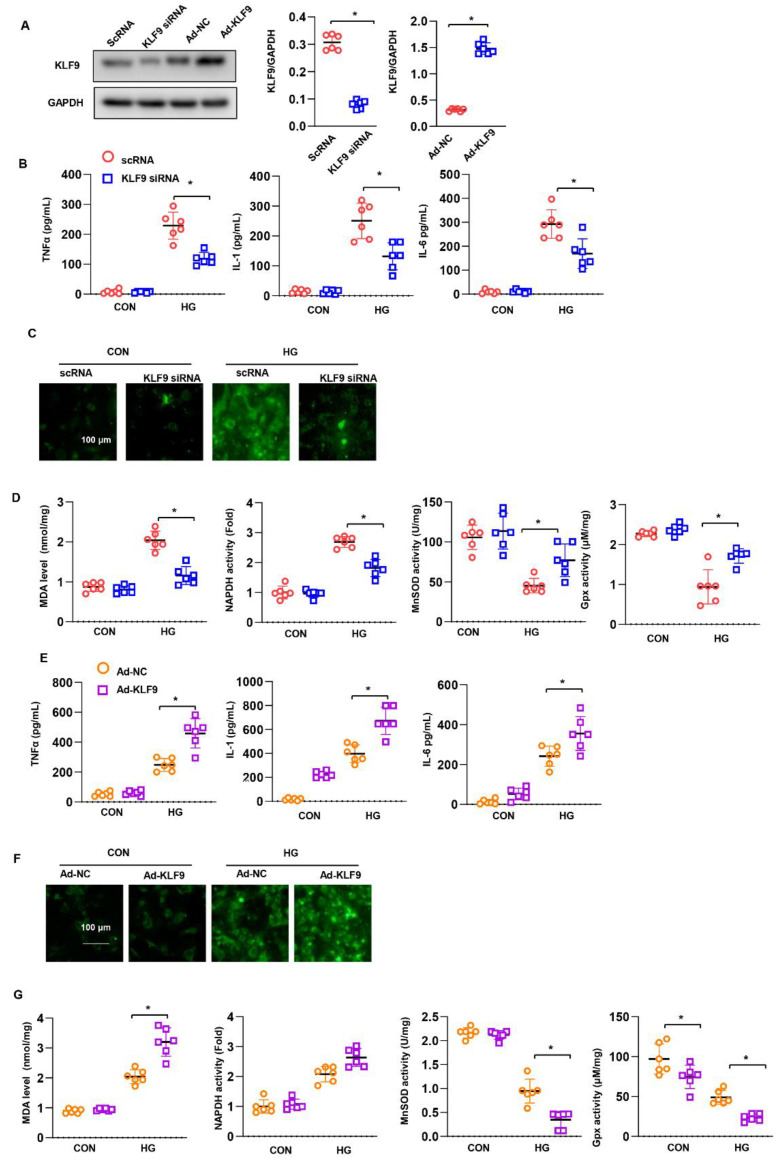
KLF9 affects cardiomyocytes in vitro. (**A**) KLF9 protein levels in cardiomyocytes transfected with Ad-KLF9 or KLF9 siRNA (*n* = 6). (**B**–**D**) Cardiomyocytes were transfected with KLF9 siRNA and treated with HG. (**B**) Cytokine levels in cells, as detected by ELISA (*n* = 6). (**C**) ROS levels (*n* = 6). (**D**) MDA levels and NADPH oxidase, SOD2 and Gpx activity in cells (*n* = 6). * *p* < 0.05. (**E**–**G**) Cardiomyocytes were transfected with Ad-KLF9 and treated with HG. (**E**) Cytokine levels in cells, as detected by ELISA (*n* = 6). F ROS levels (*n* = 6). (**G**) MDA levels and NADPH oxidase, SOD2 and Gpx activity in cells (*n* = 6). * *p* < 0.05.

**Figure 7 cells-11-03393-f007:**
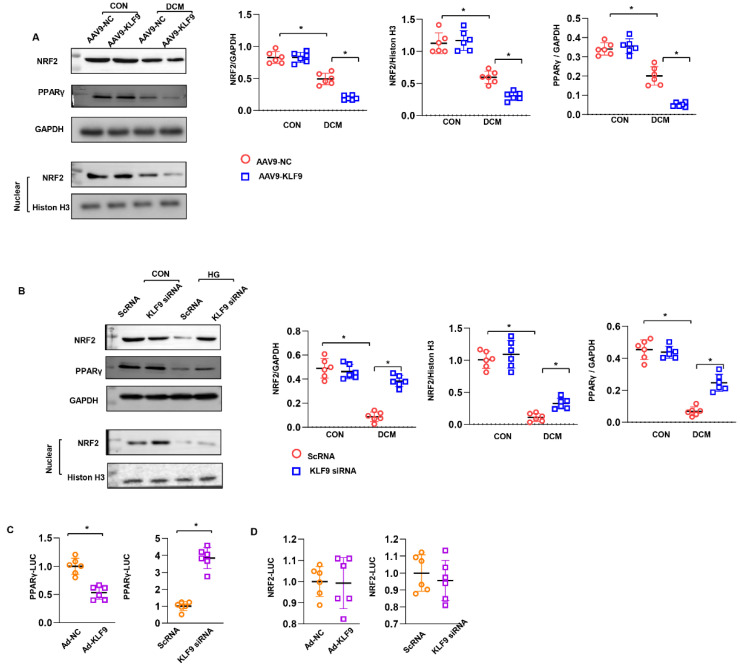
KLF9 regulatesPPARγ/NRF2 signaling. (**A**) Total and nuclear NRF2 protein levels and total PPARγ levels in mice heart with AAV9-KLF9 injection (*n* = 6). (**B**) Total and nuclear NRF2 protein levels and total PPARγ levels in cardiomyocytes transfected with KLF9 siRNA (*n* = 6). (**C**) Luciferase assay for PPARγ. (**D**) Luciferase assay for NRF2. * *p* < 0.05.

**Figure 8 cells-11-03393-f008:**
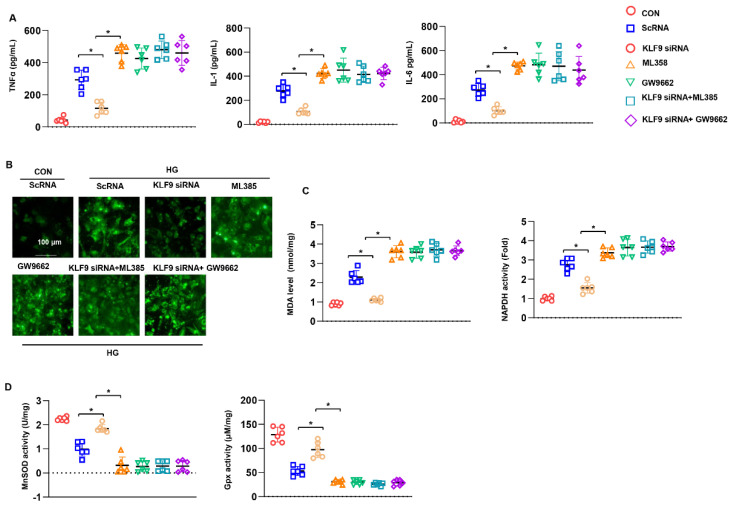
PPARγ/NRF2 inhibition counteracted the protective effects of KLF9 knockdown. (**A**–**D**) Cardiomyocytes were transfected with KLF9 siRNA, treated with the PPARγ inhibitor GW9662 (10 μM), or an NRF2 inhibitor (ML385, 1 μM) for 12 h and were then exposed to HG for 48 h. (**A**) Cytokine levels in cells, as detected by ELISA (*n* = 6). (**B**) ROS levels (*n* = 6). (**C**) MDA levels and NADPH oxidase activity (*n* = 6). (**D**) SOD2 and Gpx activity in cells (*n* = 6). * *p* < 0.05.

**Figure 9 cells-11-03393-f009:**
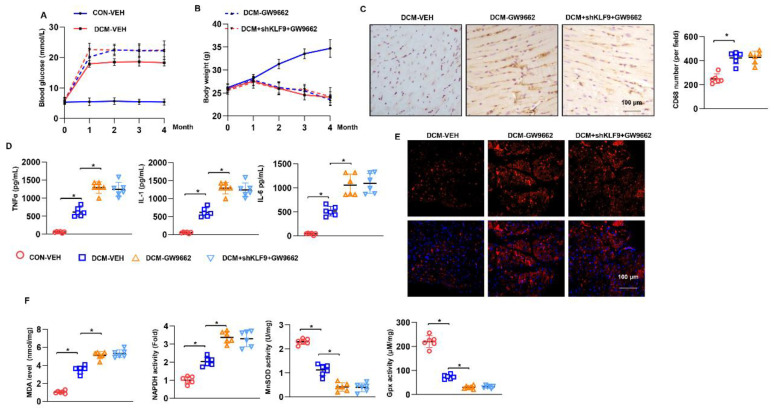
Cardiac dysfunction persists in PPARγ inhibition mice with KLF9 knockdown. DCM mice were subjected to AAV9-shKLF9 injection, and/or treated with PPARγ inhibitor GW9662. (**A**) Blood glucose in DCM mice at the indicated time points (*n* = 12). (**B**) The body weight of DCM mice at the indicated time points (*n* = 12). (**C**) CD68 staining and quantitative results in DCM mice (*n* = 5). (**D**) Cytokine levels in heart tissue in DCM mice (*n* = 6). (**E**) DHE staining in DCM mice (*n* = 5). (**F**) MDA levels and NADPH oxidase activity in DCM mice (*n* = 6). (**E**) SOD2 and Gpx activity (*n* = 6). * *p* < 0.05.

**Table 1 cells-11-03393-t001:** Primers used for real-time PCR.

Primer Name	Forward Primer	Reverse Primer
KLF9-Mouse	GCACAAGTGCCCCTACAGT	TGTATGCACTCTGTAATGGGCTTT
KLF9-Rat	GTTTGCCCCTGTAAGTAGTAAGTG	GGTTCAGGCCATTGTGTAGAC
*β-MHC*-Mouse	CCGAGTCCCAGGTCAACAA	CTTCACGGGCACCCTTGGA
*α-MHC*-Mouse	GTCCAAGTTCCGCAAGGT	AGGGTCTGCTGGAGAGGTTA
Collagen I-Mouse	AGGCTTCAGTGGTTTGGATG	CACCAACAGCACCATCGTTA
Collagen III-Mouse	CCCAACCCAGAGATCCCATT	GAAGCACAGGAGCAGGTGTAGA
*GAPDH*-Mouse	ACTTGAAGGGTGGAGCCAAA	GACTGTGGTCATGAGCCCTT
*GAPDH*-Rat	GACATGCCGCCTGGAGAAAC	AGCCCAGGATGCCCTTTAGT

## Data Availability

All data that support the findings in this study are available from the corresponding author upon reasonable request.

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
