# Peer review of "KLF9 Aggravates Streptozotocin-Induced Diabetic Cardiomyopathy by Inhibiting PPARγ/NRF2 Signalling"

_cells, 2022, doi:10.3390/cells11213393_

Round 1

Reviewer 1 Report

This is an interesting and worthy study.  I have a number of concerns about the experimental design and data presentation, as well as the MS itself.

1) The authors need to better incorporate/consolidate the literature on KLF9/oxidative stress and KLF9/cytokine synthesis/secretion into the interpretation of their own results.  They give short shrift to this and present a very minimal bibliography.  There is a much more extensive literature than cited, which along with the current study, point to a clear CONTEXT-dependent role and divergent actions of KLF9 in this regard. 

2)  It is difficult to reconcile the minimal changes (if at all) of KLF9 protein abundance with the much more pronounced effects on downstream indices.  

3) The KLF9 Western blots show no calculated mol weight for KLF9?  Were these done on nuclear extracts? How do authors know the band is actually KLF9?  

4) It appears that reasonable numbers of animals were used, yet it is unclear what a given lane in a Western blot represents?  Pool of extracts? How pooled? One representative mouse?  How were the blot data analyzed? 

5) KLF9 primer sequences are missing from Table 1?

6) In general, the IHC and whole mount images and corresponding data are difficult to see (and therefore evaluate).

7) Data corroborating significant up-regulation of KLF9 with AAV9 delivery are unconvincing; again see comment #2 above.

8) Regarding effects of KLF9 on PPARgamma, oxidative stress and cytokine synthesis, see the paper by Brown et al, Cancers, 2022, 14, 1737.

9) The Discussion requires a rewrite to better incorporate the current findings into the larger field.  In its current form, it is mostly a recapitulation of the Results. 

Reviewer 2 Report

Manuscript highlights impact of KLF9 in the diabetic cardiomiopathy, focusing on the involvement of NRF2 and disturbed oxidative balance. This manuscript could be of value to the research community; however, there are several issues that authors should address prior to the acceptance.

Please provide the exemplary images how the US analysis was done.

After dehydration, antigen repair at high temperature and high pressure, and blocking with 8% goat serum - please provide details and make sure the decription is correct

Please provide the proof about the specificity of the antibodies (Fig. 1F) - the staining of KLF9 does not seem to have exclusively nuclear localisation, whereas actin does not form fibers and seems to be mainly in the nucleus

3C - why the DHE staining is weaker in AAV-KLF9 group? General remark: how the authors explain the very weak signal of DHE during the first manipulations of KLF with AAV9?

Similarly, do what extend can KLF9 regulate the process if it is expressed in the scarcity of cells as detected by IF? How do you relate such low expression by IF to rather strong signal on IHC (Fig. 1C) - please provide negative controls

page 11, last sentence - These data indicate that KLF9 regulates NRF2 expression by directly targeting PPAR. - please explain how you infered that and if true, what could be the mechanism.

It would be useful to consider different action of NRF2, independent of its transcriptional activity. 

Round 2

Reviewer 1 Report

The authors have responded appropriately to my previous comments.

Author Response

Thanks to the valuable suggestions provided before. Your suggestions have brought lots of benefits and help to our paper.

Reviewer 2 Report

The manuscript has improved. In the first remark I meant ultrasound (US). As my remark was hard to understand and still the other questions have been answered. However, I reccomend including the description of ultrasound image analysis to increase the reproducibility of the results.

Author Response

Thanks for remind us this. We have added more details about US analysis in the method section.